# CONVERSATION CHRONICLES: Towards Diverse Temporal and Relational Dynamics in Multi-Session Conversations

**Jihyoung Jang    Minseong Boo    Hyounghun Kim**

Artificial Intelligence Graduate School, UNIST

{jihyoung, b.ms, h.kim}@unist.ac.kr

## Abstract

In the field of natural language processing, open-domain chatbots have emerged as an important research topic. However, a major limitation of existing open-domain chatbot research is its singular focus on short single-session dialogue, neglecting the potential need for understanding contextual information in multiple consecutive sessions that precede an ongoing dialogue. Among the elements that compose the context in multi-session conversation settings, the time intervals between sessions and the relationships between speakers would be particularly important. Despite their importance, current research efforts have not sufficiently addressed these dialogical components. In this paper, we introduce a new 1M multi-session dialogue dataset, called CONVERSATION CHRONICLES, for implementing a long-term conversation setup in which time intervals and fine-grained speaker relationships are incorporated. Following recent works, we exploit a large language model to produce the data. The extensive human evaluation shows that dialogue episodes in CONVERSATION CHRONICLES reflect those properties while maintaining coherent and consistent interactions across all the sessions. We also propose a dialogue model, called REBOT, which consists of chronological summarization and dialogue generation modules using only around 630M parameters. When trained on CONVERSATION CHRONICLES, REBOT demonstrates long-term context understanding with a high human engagement score.[1]

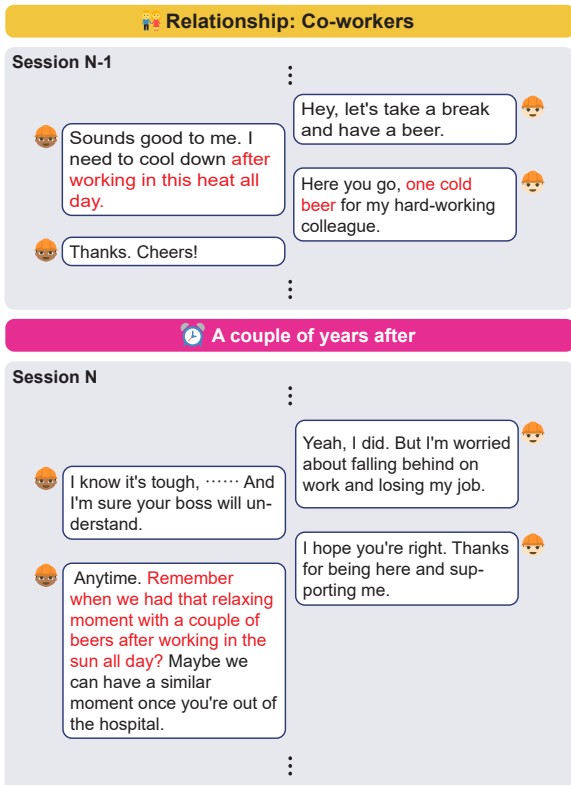

Figure 1: A sample of a multi-session conversation from CONVERSATION CHRONICLES. Based on the established relationship CONVERSATION CHRONICLES provides the relevant conversation for the user. In session N, the co-workers hold a conversation based on information remembered from previous sessions.

## 1 Introduction

Open-domain conversation is one of the important research topics. By deploying conversation systems in our daily lives, we can enjoy automated services like counseling, language tutoring, etc. There has been much research effort to build such AI conversation models (Rashkin et al., 2019; Zhang et al., 2020; Roller et al., 2021; Shuster et al., 2022). However, although these chatbot models produce human-like fluent responses, they seem to have a limited ability that only understands short-term dialogue context, making them less applicable in real-world scenarios in which long-term conversational situations are often encountered. Specifically, they do not care about the context of past conversations and only generate responses based on an ongoing dialogue (so-called single-session dialogue).

To address these issues, the multi-session con-

---

[1]Our data/code are publicly available at https://conversation-chronicles.github.io/

versation has been proposed (Xu et al., 2022a). Multi-session conversation comprises consecutive sessions that make a coherent dialogue episode. In a multi-session conversation, each session is assumed to occur serially with a time interval in between. Time interval plays an important role to infuse dynamics in a conversational interaction between speakers. For instance, depending on the time elapsed since the last conversation, their responses about past events would vary. However, previously introduced works have a relatively short range of time intervals, limiting types of transitions from the past sessions. Also, to our best knowledge, there is no research effort to incorporate the relationship between speakers into conversations. The relationship can significantly rule the way they perceive and interact with each other, giving an additional dimension of dynamics to a dialogue.

Therefore, we introduce CONVERSATION CHRONICLES, a new high-quality long-term conversation dataset that consists of 1M multi-session dialogues (200K episodes; each episode has 5 dialogue sessions). CONVERSATION CHRONICLES features more diverse chronological context and fine-grained speaker relationships. Time interval in CONVERSATION CHRONICLES includes varying ranges from a few hours to even years, allowing to cover a longer elapsed time than previous multi-session dialogue settings. Also, various relationships induce varied events and interaction flows to the conversations, which facilitates the application to different real-world scenarios (Figure 1).

On the other hand, collecting data samples, which requires sophisticated interaction, at scale is not easy and time-consuming. Thus, recent works are getting turning to exploit large language models (LLMs) to collect such complicated data in an automated way by designing refined query methods (Kim et al., 2022a; Taori et al., 2023; Xu et al., 2023; Zheng et al., 2023; Gilardi et al., 2023). Following those works, we collect our multi-session conversation dataset through well-defined prompts to LLMs.[2] To be specific, each prompt consists of relationships, event descriptions, and time intervals so that the created dialogues incorporate those components. According to human evaluation based on multiple criteria, our CONVERSATION CHRONICLES is preferred to other multi-session conversation datasets.

We also propose a new multi-session conversation model, REBOT. This model uses only about 630M parameters and reflects the chronological and relational dynamics in the long-term conversation setting. The extensive human evaluation shows that its responses are preferred over other chatbot models in long-term conversational situations.

Our contributions in this study are:

1. We introduce CONVERSATION CHRONICLES, a new 1M multi-session dataset that includes more various time intervals and fine-grained speaker relationships.

2. We propose REBOT which can generate dialogues with the chronological dynamics with only about 630M parameters.

3. Extensive human evaluation verifies that RE-BOT trained on CONVERSATION CHRONICLES shows user engagement in situations with various temporal and relational contexts.

## 2 Related Works

**Open-domain Chatbot.** Building human-like open-domain dialogue models is an important research topic in the field of natural language processing. Diverse datasets have been proposed to study such chatbots. Previous studies of open-domain dialogue datasets are DailyDialog (Li et al., 2017), PersonaChat (Zhang et al., 2018), Wizard of Wikipedia (Dinan et al., 2019), Empathetic Dialogues (Rashkin et al., 2019), Blended-SkillTalk (Smith et al., 2020), twitter (Ritter et al., 2011) and Pushshift.io Reddit (Baumgartner et al., 2020). However, these dialogue datasets consist of short, single sessions, making it difficult to reflect real-world conversational scenarios in which conversations occur in series with time intervals.

**Long-term Conversation.** Current open-domain dialogue models learn from short conversations with little context, which has the obvious limitation that they will not remember the information for future conversations. To address these issues, there are attempts to add modules to the standard architecture or propose datasets for long-term situations. Wu et al. (2020) proposes a method of extracting and managing user information from dialogues. Xu et al. (2022b) proposes a Chinese multi-turn dataset DuLeMon and a persona memory-based framework PLATO-LTM. Xu et al. (2022a) proposes the first multi-session dataset, called MSC, with time intervals between sessions. Also, Bae

---

[2]We use ChatGPT (OpenAI, 2022) in this study, but other LLMs, like Bard (Google, 2023), could also be employed.

et al. (2022) proposes a dynamic memory management method to keep user information up-to-date and introduces a Korean multi-session dialogue dataset, CareCall$_{mem}$. However, previous multi-session datasets have a limited or fixed range of time intervals and they have less focused on the impact of the time interval in training dialogue models. Also, there is still no open-domain dialogue dataset that constructs conversations taking into account the relationship between speakers, which is quite important for engaging conversation experiences. To our best knowledge, CONVERSATION CHRONICLES is the first open-domain dialogue dataset that defines the fine-grained relationships between speakers with a diverse range of time intervals.

**Data Distillation.** Data collection is one of the most challenging problems in training AI models. This is due to copyright and privacy issues, as well as the high cost of hiring humans to generate high-quality data. Since the large language models emerged, researchers have been trying to solve the data collection problem by using them. Zheng et al. (2023) uses GPT-J (Wang and Komatsuzaki, 2021) to generate an emotional dataset, AugESC. Through an augmentation framework using GPT-3 (Brown et al., 2020), Kim et al. (2022b) created ProsocialDialog, a dataset that teaches conversational agents to respond to problematic content based on social norms. Kim et al. (2022a) uses InstructGPT (Ouyang et al., 2022) to create dialogues from narratives. Xu et al. (2023) demonstrates creating a single-session dataset using ChatGPT (OpenAI, 2022). These studies suggest that building datasets using LLMs can save time and cost, and also enable the creation of high-quality datasets that are comparable to human-written ones (Gilardi et al., 2023). Therefore, we also leverage LLMs to efficiently build the large-scale multi-session dataset, CONVERSATION CHRONICLES.

## 3 CONVERSATION CHRONICLES

In this study, we introduce a new high-quality long-term multi-session conversation dataset, called CONVERSATION CHRONICLES. Our dataset consists of 1M multi-session dialogues, comprising a total of 200K episodes, each of which consists of 5 sessions. We construct our dataset using the following process.

### 3.1 Event Collection

In a single-session dialogue, two speakers engage in a conversation around a specific event ignoring any past context. On the other hand, in a multi-session dialogue, the context of previous sessions is taken into account and reflected in the conversation of the ongoing session. This ensures coherence and continuous conversational experience in long-term conversations by preserving the context of the entire sessions.

Therefore, when creating multi-session dialogues, it is important to keep a consistent and coherent context throughout an episode. To guarantee this, we build an event graph by linking related events. To be specific, we employ the narratives from SODA (Kim et al., 2022a),[3] which is one of the large-scale dialogue datasets, and use them as the events (i.e., one narrative corresponds to an event). Then, we connect each event to another based on their relevance and build them into a graph as follows.

**Event Pairing.** We use natural language inference (NLI) as the method for linking two related events since it is one of the most reliable ways to model relationships between sentences. An event pair is classified as entailment, neutral, or contradiction, depending on whether they are related or not. We compute the relationship between all possible event pairs and retain only ones that have the entailment relationship. We employ the pre-trained BERT-base model (Devlin et al., 2019) and fine-tune it on the SNLI (Bowman et al., 2015) corpus.

**Event Graph Building.** Since a graph is an effective structure for modeling relationships between nodes, we conceptualize events as nodes and connect the entire event pairs using edges. To prevent temporal contradiction between events, we use a directed graph by which the order of premises and hypotheses is specified. From the graph, we extract all possible event sequences with a length of 5, then remove ones if they have more than 3 events in common, leaving only one of them in the list.

### 3.2 Chronological Dynamics

CONVERSATION CHRONICLES integrates diverse temporal contexts and fine-grained speaker relationships in multi-session conversations to implement chronological dynamics. Unlike single session one,

---

[3]Although we use the SODA's narrative in this study, any event descriptions could also be used.

| Time Interval | MSC (Xu et al., 2022a) | CONVERSATION CHRONICLES |
|---|---|---|
| A few hours | 3,497 | 159,975 |
| A few days | 3,510 | 159,928 |
| A few weeks | - | 160,670 |
| A few months | - | 160,050 |
| A couple of years | - | 159,377 |

Table 1: Statistics of the time interval between sessions of MSC and our dataset. We aggregate 1-7 hours as a few hours and 1-7 days as a few days in MSC.

| Relationship | Count | Ratio |
|---|---|---|
| Classmates | 66,090 | 33.05% |
| Neighbors | 49,521 | 24.76% |
| Co-workers | 28,856 | 14.43% |
| Mentee and Mentor | 16,035 | 8.02% |
| Husband and Wife | 13,486 | 6.74% |
| Patient and Doctor | 6,980 | 3.49% |
| Parent and Child | 6,514 | 3.26% |
| Student and Teacher | 5,018 | 2.51% |
| Employee and Boss | 4,811 | 2.41% |
| Athlete and Coach | 2,689 | 1.34% |
| **Total** | **200,000** | |

Table 2: Statistics of the relationship between speakers in CONVERSATION CHRONICLES.

a multi-session conversation considers previous sessions, having a time interval between each consecutive session pair. While previous studies have employed the time interval between sessions, the interval typically ranges from a few hours to several days, only allowing for a relatively short-term conversational context. Also, there has been no prior effort to apply the relationships between speakers to conversational interactions, thus limiting the variety and leading to monotonous interactions.

**Time Interval.** To address the short-term interval limitation and allow for longer dynamics, we define a longer chronological context ranging from a few hours to a few years: "a few hours later", "a few days later", "a few weeks later", "a few months later", and "a couple of years later". We randomly pick one and assign it as a time interval for a consecutive session pair. We employ approximate time representations (i.e., 'a few' or 'a couple of') rather than a numerical time amount (e.g., '3 days') since we find that minute differences in time units have little effect on the context. Please refer to Table 1 to see the comparison.

**Speaker Relationship.** We define a fine-grained speaker relationship for each episode to give interactional dynamics to a dialogue. Relationships between speakers are one of the crucial elements of dialogue since it determine the contents that they are speaking about. Since relationships are closely

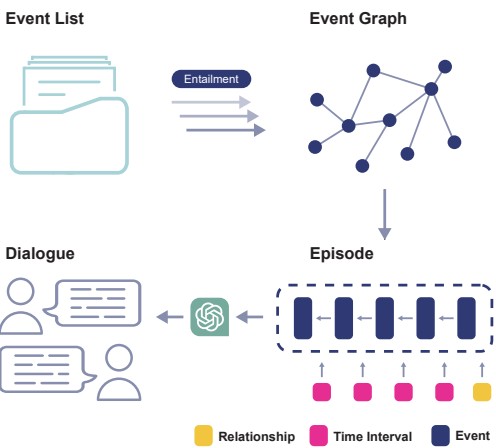

Figure 2: The overall data collection process of CONVERSATION CHRONICLES.

connected to the dialogue context (i.e., events), it would not be appropriate to assign them randomly. Therefore, we pre-define 10 relationships that are typically found in our daily lives and assign them by querying ChatGPT. To be specific, we provide all events in an episode with the list of 10 relationships, then ask ChatGPT to select the most appropriate relationship for the events. Please see Table 2 for the frequency of the relationships and Appendix A for the prompts used.

### 3.3 Conversation Episode Generation

LLMs are reported to be able to produce diverse and high-quality data samples that are comparable to those written by humans (Gilardi et al., 2023). Recent works have also reported the use of LLMs to collect dialogues (Kim et al., 2022a; Xu et al., 2023). Thus, we leverage LLMs to generate dialogues. Specifically, we collect episodes through ChatGPT (OpenAI, 2022) by designing a series of sophisticated prompts. One prompt for a session consists of an event, a time interval, and a speaker relationship as the conditions of the current dialogue, while also containing the full context (events and time intervals of previous sessions). Please refer to Appendix A for examples of the full prompts.

Using the prompts, we construct a large-scale multi-session conversation dataset, CONVERSATION CHRONICLES. By integrating the aforementioned ingredients (events, time intervals, and relationships), it implements chronological dynamics making the multi-session conversation setting more diverse. Please see Figure 2 for the overall process of the data collection.

We collect a total of 200K episodes each of which has 5 sessions, resulting in 1M dialogues.

| Datasets | Language | # of Sessions | # of Episodes | # of Turns | Avg. Turns per Session | Avg. Words per Turn |
|---|---|---|---|---|---|---|
| MSC (train, up to 3 sessions) | English | 12K | 4K | 161K | 13.5 | - |
| MSC (train, up to 4 sessions) | English | 4K | 1K | 53K | 13.3 | - |
| CareCall$_{mem}$ | Korean | - | 7.7K | 160K | 20.9 | 4.93 |
| CONVERSATION CHRONICLES | English | 1M | 200K | 11.7M | 11.7 | 18.03 |

Table 3: Comparison of ours with other multi-session datasets. Statistics for MSC and CareCall$_{mem}$ are taken from their papers. As we can see, our CONVERSATION CHRONICLES has the largest scale. On the other hand, the average of turns per session is smaller than other datasets, but the average of words per turn is much higher.

Please see Table 3 for more statistics of our CON-VERSATION CHRONICLES dataset. As the statistics show, we build a significantly larger multi-session conversation set than the others. Please see Appendix H for the full dialogue episodes.

### 3.4 Quality

**Automatic Filtering.** Data generated by an LLM does not always guarantee uniform quality. It may include unnecessary information or deviate from the given format. To ensure the consistent quality of our dataset, we implement an automatic process to filter out such cases (please see detailed process in Appendix B). Furthermore, our dataset might have the potential to contain toxic data. Thus, we use Moderation (Markov et al., 2023) to remove the harmful data.

**Human Evaluation.** We conduct human evaluations to verify the quality of our CONVERSATION CHRONICLES (see Section 5 for the evaluation details). We sample 5K episodes and ask evaluators to rate each of them based on four criteria (coherence, consistency, time interval, and relationship). Table 4 shows the quality of CONVERSATION CHRONICLES, with an average score of 4.34 out of 5 which is quite high considering 5 indicates 'perfect'.

We also conduct a comparison with MSC (Xu et al., 2022a), one of the representative multi-session conversation datasets, based on the same criteria as the previous evaluation, excluding the relationship since the MSC does not have it. We randomly sample 0.5K dialogue episodes from each dataset for comparison. As shown in Figure 3, our dataset has higher scores across all metrics, meaning our CONVERSATION CHRONICLES has such high quality.

## 4 REBOT

We propose a novel multi-session dialogue model, REBOT (**RE**member Chat**BOT**). REBOT consists of two parts: the chronological summarization module and the dialogue generation module. The sum-

| Metrics | Avg | Std |
|---|---|---|
| Consistency | 4.41 | 0.80 |
| Coherence | 4.04 | 1.06 |
| Time interval | 4.46 | 0.77 |
| Relationship | 4.40 | 1.08 |
| **Overall** | **4.33** | |

Table 4: Human evaluation on the quality of CONVER-SATION CHRONICLES.

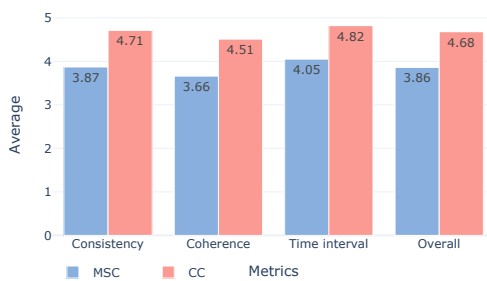

Figure 3: Comparative evaluation of MSC and CON-VERSATION CHRONICLES (CC).

marization module provides the context of the previous sessions by concisely describing the chronological events. The dialogue generation module produces the next response reflecting chronological dynamics presented by the summarization module.

### 4.1 Chronological Summary

Multi-session conversations must take into account the chronological connectivity between previous and current sessions, and appropriately reflect changes in event states over time. The best solution to soundly incorporate this information in the dialogues model would be to put the entire conversation history of previous sessions as context. However, it is not computationally efficient to maintain such a system. To address this inefficiency, we propose a summarization module that depicts each of the past dialogue sessions while minimizing information loss.

To collect training data for this summarization module, we randomly sample 100K sessions (i.e., 20K episodes) from CONVERSATION CHRONICLES and use ChatGPT to generate a summary for each session. We employ T5-base (Raffel et al., 2020), and use 80K from the generated summaries

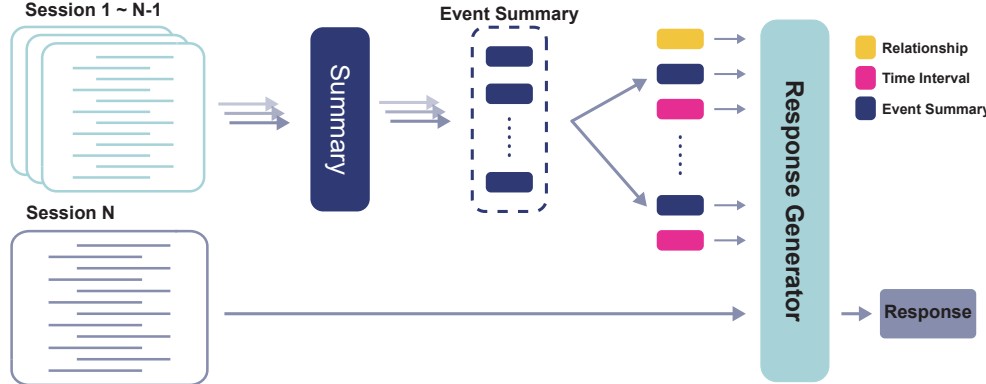

Figure 4: The overall architecture of REBOT. It consists of a summarization module and a generation module. The yellow box indicates the relationship, and the pink box indicates the time interval between dialogue sessions. The summarization module summarizes the previous sessions as input to the generation module.

for training (keeping 20K for val/test splits). The module takes session dialogue as input and generates a chronological summary as output.

## 4.2 Dialogue Generation

To generate utterances in an ongoing session, the dialogue generation module should consider the dialogue history (i.e., previous session summaries), the relationship between speakers, and the time elapsed from the last session. We use a sequence-to-sequence architecture, i.e., BART-large (Lewis et al., 2020), to effectively accommodate all the components to be considered during the generation process. Formally, the conditional probability for generating the next response is $P(c|r, t, s, h)$, where $c$ is the next utterance, $r$ is the relationship, $t$ is the time interval, $s$ is the summary, and $h$ is the current dialogue context (i.e, previously generated utterances). The input format to the module looks like: $<$relationship$>$ $r$ $<t_N>$ $s_{N-1}$ $<$user$>$ $u_1$ $<$bot$>$ $c_1$ $<$user$>$ ... $<$bot$>$ $c_n$.

The overall model architecture is shown in Figure 4. REBOT trained on CONVERSATION CHRONICLES can seamlessly generate multi-session dialogues considering chronological events and long-term dynamics only with 630M parameters.

## 5 Experiments

### 5.1 Implementation and Training Details

We split the dataset into 160K for train, 20K for validation, and 20K for test, out of 200K episodes (1M dialogue sessions). We use AdamW (Loshchilov and Hutter, 2019) as the optimizer and cross-entropy loss as the training objective for all generation tasks. Please see Appendix C for more details.

### 5.2 Human Evaluation

Evaluating the quality of open-domain conversations is considered challenging. The reference-based evaluation metrics (such as BLEU (Papineni et al., 2002), ROUGE (Lin, 2004), etc.) might not be suitable for use as evaluation metrics for open-domain dialogues, for which a wide variety of generations could be considered as proper responses (Liu et al., 2016). Therefore, human evaluation is desirable to faithfully verify the quality of dialogues on various aspects (such as coherence, contradiction, engagement, etc.). As such, in this study, we rely on extensive human evaluation for examining the quality of our dataset, CONVERSATION CHRONICLES, and dialogues generated by our REBOT.[4]

### 5.3 Dataset Quality Evaluation

CONVERSATION CHRONICLES implements the chronological dynamics in a multi-session conversation environment. To ensure that our dataset faithfully reflects the elements (events, time intervals, and relationships) in the dialogues, we randomly sample 5K episodes for evaluation, then conduct a human evaluation by asking the evaluators to rate the dialogues based on 'Coherence', 'Consistency', 'Time interval', and 'Relationship'. Please see Appendix I for the detailed definition of those criteria.

### 5.4 Comparison to Other Datasets

We also perform a human evaluation for comparison with an existing multi-session dataset. Since just a few multi-session datasets have been introduced and MSC (Xu et al., 2022a) is the only case

---

[4]We hired 41 evaluators from a professional evaluation agency and 5 in-house evaluators for the evaluation.

| Metrics | Avg | Std |
|---|---|---|
| Engagingness | 4.78 | 0.54 |
| Humanness | 4.74 | 0.63 |
| Memorability | 4.14 | 0.79 |
| **Overall** | **4.55** | |

Table 5: Human evaluation for the quality of dialogue episodes generated by REBOT.

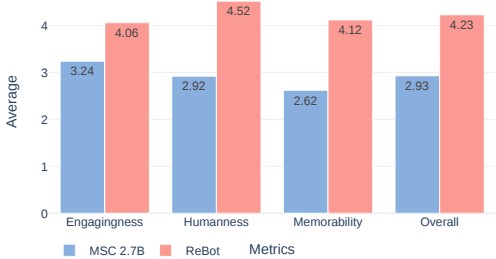

Figure 5: Comparative evaluation results of MSC 2.7B and REBOT.

for a fair comparison due to its session length and language, we choose MSC for the comparison.

We extract all possible episodes with 5 sessions from MSC (validation 0.5K, test 0.5K, total 1K), then randomly sample 0.5K episodes for this comparison. We also randomly sample 0.5K episodes from CONVERSATION CHRONICLES, excluding those previously extracted for the aforementioned quality evaluation. We use the same metrics for this comparison except for 'Relationship' since there is no relationship between speakers in MSC. For a more reliable comparison, we perform a consensus evaluation. A single episode is rated by three human evaluators, and then we average their scores and take it as the final score of the episode.

### 5.5 Model Performance Evaluation

**Summarization Performance.** We randomly sample 3K generated summaries from the summarization module (1K from each of the second, third, and fourth sessions) and ask evaluators to judge whether the generated summary fully describes the interaction between speakers in the dialogue.

**Generation Performance.** We randomly extract 1K of the first sessions (0.1K session from each of the 10 relationships). Then, we take the relationship and the summary of the first session as input to generate the second session, then keep generating the following sessions self-regressively. The time interval between sessions is randomly chosen and the dialogue continues in each session until [END] is generated. We ask evaluators to evaluate the generated 1K episodes based on 'Engagingness', 'Humanness', and 'Memorability'. Please see Appendix I for the definition of those criteria.

### 5.6 Comparison to Other Models

We also conduct interactive evaluations and compare them with another multi-session dialogue model. Since the only multi-session dialogue model that is publicly available is MSC 2.7B (Xu et al., 2022a), we choose it. We ask in-house evaluators to evaluate with the same criteria as above. Evaluators are asked to rate responses of the mod-

els by having a conversation with those models on their own (at least 6 turns), considering the persona (in the case of MSC 2.7B) or event summary (in the case of REBOT) of the previous session. For this comparison, the evaluators conduct 50 live chats with each model. Please refer to Appendix I for more details about human evaluations.

## 6 Results

In this section, we present the evaluations of our REBOT's performance in different experiment setups (please see Section 3.4 for the quality of our dataset, CONVERSATION CHRONICLES).

**Dialogue Generation.** Table 5 shows the evaluation of multi-session dialogues generated by RE-BOT. The quite high scores across all the metrics imply that each generated episode is considered natural and engaging like real human conversation. Also, it is rated to have good memory retention with little contradictions from the sessions generated earlier in the dialogue. This corresponds to the consistency factor in CONVERSATION CHRONICLES quality evaluation. Please see Appendix G for more detailed statistics by relationship.

**Interactive Dialogue Generation.** We examine the user experience of our model in an interactive dialogue setting. Figure 5 shows the comparison between our model and MSC 2.7B. The better performance of REBOT compared to MSC 2.7B should be because the characteristics of CONVERSATION CHRONICLES are well reflected in the model (i.e., coherency, consistency, awareness of time elapsed, and speaker relationship). Especially, through an informal short survey, evaluators report that having a conversation based on a pre-defined relationship is more focused than not having any relationship, proving that the fine-grained relationships introduced in our CONVERSATION CHRONICLES work effectively (see the next paragraph for detailed evaluation for the role of 'relationship').

| Relationship: Husband and Wife |
| --- |
| **USER:** I think my salary is too low for my workload. |
| **REBOT:** What do you mean? |
| **USER:** I started a new project these days. So I work overtime a lot, but I don't think the compensation is appropriate. |
| **REBOT:** Have you talked to your boss about it? |
| **USER:** No, not yet. Honey, what do you think I should do? |
| **REBOT:** Well, you should definitely talk to your boss and explain your situation. Maybe they can give you a raise. |
| Relationship: Employee and Boss |
| **USER:** I think my salary is too low for my workload. |
| **REBOT:** I understand where you're coming from, but we need to be realistic about our budget. Can you give me some specific examples of why you feel that way? |
| **USER:** As you know, I've been assigned a new project and the workload has doubled. I work a lot of overtime, but I think my salary is low. |
| **REBOT:** I see your point, but we also have to consider the company's financial situation. |
| **USER:** Then can you give me a bonus if the project ends successfully? |
| **REBOT:** That's a good idea. Let's discuss it further and see if we can come up with a solution that works for both of us. |

Table 6: An example of the dialogue flows based on relationships. REBOT generates different responses depending on relationships.

| Relationship: Athlete and Coach |
| --- |
| **USER:** Actually, I've been thinking about it since my last injury, but now I'm going to retire as an athlete and live a new life. |
| ...... |
| **USER:** Thank you. I will probably take coaching training for a year, and I am planning to be a full-time coach after that. It would be meaningful to meet each other as a coach later on. |
| **REBOT:** That sounds like a great plan. I wish you all the best in your coaching career, Athlete. |
| A couple of years after |
| **USER:** Hey, I finished my training course and finally became a full-time coach like you! |
| **REBOT:** That's great news! I knew you had it in you. |
| **USER:** Thank you! **As I became an official coach**, I'd like to learn what I need to know as a coach, could you tell me? |
| ...... |
| **REBOT:** Exactly. **As a coach, you need to understand your players and their motivations**. You need to be able to connect with them on a deeper level and help them reach their full potential. |
| **USER:** Thank you! I will be a coach who can raise the potential of the players like you! |
| **REBOT:** I have no doubt that you will. **Just remember to stay true to yourself and your values as a coach**, and always lead by example. |

Table 7: An example of shifts in relationships. When the relationship is shifted in the last session, REBOT can recognize it.

This demonstrates that our model effectively learns chronological dynamics from the dataset. Please see Appendix H for examples of the user interactions with REBOT and MSC 2.7B.

**Speaker Relationship.** Table 6 shows an example of dialogue flow that varies by a different relationship for the same conversational topic. As we can see, a dialogue beginning with a similar context can lead to different interactions depending on speaker relationships. This means that defining a relationship allows conversations to have varying levels of expression such as emotional depth and information exchange. Please see Appendix D for examples of more speaker relationships.

Additionally, Table 7 shows an example of relationship shift across sessions. The dialogue begins with an initial relationship between 'Athlete and Coach'. However, USER (the Athlete) decides to be a coach, transitioning to work alongside their former coach as peers. REBOT (the Coach) can recognize the change in relationship and respond to the conversation accordingly. This shows that our model can handle shifts in relationships due to events and the passage of time, as our dataset incorporates both temporal and relational dynamics.

**Time Interval.** Table 8 shows the effects of time intervals in a dialogue episode. The example shows that an event that occurred in a past session is recalled in the following session assuming a given time has elapsed. In particular, we can see that it has a memory for past events, even if they are not in the immediately preceding session, and it correctly reflects the accumulation of time intervals.

**Chronological Summarization.** We conduct human evaluations to check the quality of the gener-

| |
|---|
| **Relationship:** Student and Teacher |
| ...... |
| **USER:** Why don't we go the beach in front of our school? |
| ...... |
| **A few weeks after** |
| ...... |
| **USER:** Wow, it's already vacation! I had a lot of fun this semester. |
| ...... |
| **A couple of years after** |
| ...... |
| **USER:** I can't believe I'm graduating already. Thank you so much, teacher. |
| ...... |
| **A few hours after** |
| ...... |
| **USER:** What's the most memorable memory you had with us? |
| **REBOT:** Oh, that's a tough one. I think the most memorable memory was when we all went to the beach together. |
| **USER:** Yes, it was fantastic. It's a really old memory. |
| **REBOT:** Speaking of old memories, do you remember when you went on vacation a few years ago? |
| ...... |

Table 8: An example of dialogue between REBOT and a user over multiple time intervals. This example shows the model can recall past session events considering the time intervals.

ated summaries. The average score of the total of 3K samples is 4.3 out of 5, indicating the summarization module works well enough to support the generation module by providing important context from previous sessions. Please see Appendix E for examples of chronological summary.

**Ablation Study.** We perform ablation experiments to ascertain the significance of incorporating both time and relationship information in our model. When the model is trained devoid of time interval data, it exhibits a trend toward producing responses with generic time information. In the absence of relationship information during the training phase, the model fails to uphold a consistent relationship context with the user. For illustrative examples, please refer to Appendix F.

## 7 Conclusion

We introduce a large-scale multi-session conversation dataset, CONVERSATION CHRONICLES, which implements chronological dynamics by integrating time interval and speaker relationship in it. To create the multi-session conversations, we first build an event graph and then distill a series of dialogues from ChatGPT using well-defined prompts based on the event graph, time interval, and speaker relationship. We verify the quality of our dataset with extensive human evaluations on diverse metrics and criteria. We also propose REBOT, a multi-session dialogue model, which comprises chronological summarization and dialogue generation modules. The results of human evaluations show our REBOT can generate diverse coherent responses according to different time intervals and speaker relationships with high user engagement without contradiction in a long-term conversation setup.

## Limitations

We focused on developing an interactive dialogue model that reflects time intervals and relationships. However, the research was conducted with a limited number of specific time intervals and speaker relationships. This limitation could potentially limit the generalizability of the research findings. In the future, we plan to expand the research to include more time intervals and speaker relationships.

In addition, the choice of LLMs can significantly affect the type of dialogue generated. In other words, using different LLMs could lead to different results and types of dialogues, even when using the same framework. Therefore, we plan to consider configurations that mix different LLMs as a valuable resource for generating different types of dialogue and content.

## Ethics Statement

Despite our best efforts, potentially harmful content may be included in the data. Although our model is trained on a toxic-filtered dataset, it may generate responses that users do not want. In addition, the responses generated by our model might not be applicable in the real world. For example, medical advice given by a model could not be appropriate in a real medical situation. We recommend using our model for research purposes, or with care in

real-world applications. Additionally, we communicated with the evaluation agency to check that the annotators were being compensated fairly.

## Acknowledgements

We thank the reviewers for their valuable feedback. We thank Seokhyun An and Jaeho Oh for their helpful discussion. This work was supported by Institute of Information & communications Technology Planning & Evaluation(IITP) grant funded by the Korea government(MSIT)(No.2020-0-01336, Artificial Intelligence Graduate School Program(UNIST)) and the 2022 Research Fund (1.220140.01) of UNIST(Ulsan National Institute of Science & Technology).

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

## A  Prompts Details

**Prompts for Relationship.**  We use ChatGPT to assign a fine-grained speaker relationship to each episode.  The prompt is used as follows: "Two people want to have a conversation about the topic below. Please choose from the options below the most appropriate relationship between the two speakers in the conversation. Don't recommend other options.  You are responding without comment.  Also, your answer is limited to the options below.\n\nTopic: {Episode Event Description}\n\nOption:\n1. Husband and Wife\n2.  Child and Parent\n3. Co-workers\n4.  Classmates\n5.  Student and Teacher\n6. Patient and Doctor\n7. Employee and Boss\n8.  Athlete and Coach\n9.  Neighbors\n10. Mentee and Mentor"

**Prompts for Conversation.**  We use ChatGPT to collect multi-session dialogues for CONVERSATION CHRONICLES.  The prompt is used as follows:  "The following is a next conversation between {Relationship}.\n\nThe {Relationship} took turns talking about the below topics:\n{Session N-1 Event Description}\n\n{Time Intervals Between Session N-1 and N} the last topic, this is the topic {Relationship} are talking about today:\n{Session N Event Description}\n\n{Speaker A}'s statements start with [Speaker A] and {Speaker B}'s statements start with [Speaker B]. {Speaker A} and {Speaker B} talk about today's topic, and if necessary, continue the conversation by linking it to the conversation topic of the past. Complete the conversation in exactly that format."

**Prompts for Summary.**  We use ChatGPT to generate chronological event summaries.  The prompt is used as follows:  "You're a summarizer.  Choose the most important events from a given conversation and summarize them

in two sentences.\n\n[Conversation]\n\nSession Dialogues\n[Summary]"

**Prompts Environments.**  ChatGPT uses reinforcement learning from human feedback. We use the "gpt-3.5-turbo-0301" model to ensure reproducibility, as responses may be different depending on the version.

## B  Dataset Filtering Process

To ensure uniform quality of CONVERSATION CHRONICLES, we filter out the following cases: (1) sessions with more than two speakers; (2) sessions with unclear alignment between utterance and speaker; (3) sessions where speakers not included in the pre-defined relationship appear; (4) sessions with unnecessary information such as stage directions (e.g., any descriptions of actions or situations). We remove all conversation episodes that include at least one of these cases.

## C  Implementation and Training Details

**Summarization Module.**  We employ the pre-trained T5-base model consisting of 222M parameters for the summarization module. We train with the linear scheduler, a batch size of 32, 512 for the maximum length of the input sequence, and 128 for the maximum length of the output sequence. Training takes about 6 hours on 8 NVIDIA RTX A6000 GPU devices with a maximum of epoch 5 with early stopping.

**Generation Module.**  We use the pre-trained BART-large model consisting of 406M parameters for the generation module. We train with the linear scheduler, a batch size of 16, 1024 for the maximum length of the input sequence and 128 for the maximum length of the output sequence. Training takes about 3 days on 8 NVIDIA RTX A6000 GPU devices with a maximum of epoch 3 with early stopping.

All pre-trained models used are based on Hugging Face Transformers (Wolf et al., 2020).

## D  Dialogue Example for Each Relationship

Our CONVERSATION CHRONICLES and REBOT incorporate fine-grained relationships in a multi-session environment. There are 10 relationships in total, and the model response differently depending on the relationship, even for conversations about the same context. Table 9 shows different dialogues

for the same context for each relationship. In the dialogue, the user talks about the difficulty of homework in various relationships. The session begins with the same utterance for each relationship, but the flow of the dialogue varies depending on the relationship. As we can see in the example, the classmate advises their friends to ask the teacher for help, and the teacher opens a supplementary class. Parents, also provide emotional support to the child. In other words, the same context can obtain different responses depending on the relationship, such as providing advice or giving empathy. This demonstrates the works of our fine-grained relationship.

Also, Table 10 to 19 shows example dialogues for all relationship options. The examples are dialogues between a user and REBOT, and we can see that defining a relationship works quite well. To our best knowledge, our work is the first to integrate relationships into dialogues, and extensive human evaluation shows that the relationship between the speakers helps to achieve high engagement in the conversation.

## E   Chronological Summary Example

Please see Table 20 for a chronological summary example. In the previous session, the coach and the athlete were scheduled to start training in the morning. However, after a few hours, the coach and the athlete decide to change their training schedule. The chronological summary effectively captures the state changes of these events, detailing the shift in the training schedule from morning to afternoon. Next, in Table 21, husband and wife spend time in parks and restaurants on their wedding anniversary in previous sessions. After a few months, they decide to have dinner at a restaurant. The husband offers a meal at the restaurant they went to on their wedding anniversary. Then, they have a conversation recalling memories at the restaurant. We can see that the summary accurately reflects theirs reminisces.

## F   Ablation Study Example

We incorporate temporal and relational dynamics in REBOT. To verify the impact of these dynamics, we conduct ablation experiments. Table 22 shows an ablation study example to assess the impact of time information in REBOT. This example suggests that a model trained with time information can produce responses that are specific to the time interval. In contrast, a model trained without time information generates more generic time-related responses. This implies that a model trained with time information is better at capturing and incorporating specific time intervals into its responses. Next, Table 23 shows another ablation study example that focuses on the influence of relationship modeling. This example indicates that a model trained with relationship information produces more contextually consistent responses. In contrast, considering the model's response, a model trained without relationship information exhibits contextually inconsistent responses. This suggests that models trained with relationship information can better understand the context and generate responses that align with the nature of the relationship. Overall, it appears that incorporating time interval and relationship into REBOT can improve its ability to generate contextually appropriate and specific responses.

## G   Generation Quality per Relationship

Table 24 shows the evaluation scores of generated dialogue by REBOT for each relationship category. As we can see, the scores are balanced across all relationships, meaning that REBOT effectively mirrors relational dynamics for all relationships.

## H   Episode Example

Figure 6 shows a sample of the entire episode of CONVERSATION CHRONICLES. Figure 7 shows an example of a live chat between a user and REBOT. Figure 8 shows a live chat sample for one session with MSC 2.7B.

## I   Human Evaluation

We conduct human evaluations by asking the evaluator to read the full episode and rate the dialogue quality based on the following metrics. The score of each metric ranges from 1 to 5 with 5 meaning perfect for a corresponding metric (Section 5.3 and Section 5.4).

- **Coherence**: The conversation between two speakers should have a natural flow in terms of event transition.
- **Consistency**: Two speakers should not make a contradiction from past sessions.
- **Time interval**: The speakers should make a conversation in each session as if the designated time has elapsed since the last session.

- **Relationship**: Two speakers are having a conversation with the designated relationship. Throughout the session, the two speakers must maintain this relationship.

We ask evaluators to evaluate model performance based on following three criteria (Section 5.5 and Section 5.6).

- **Engagingness**: Two speakers should interact to create responses that are not only interesting but also well-immersed in the given context of the conversation.
- **Humanness**: Two speakers should have a conversation that demonstrates emotional understanding (e.g., empathy) and the use of natural language and thought processes that are typical of human beings.
- **Memorability**: If two Speakers recall past events correctly by retaining information from previous sessions.[5]

Evaluators conduct their evaluations on the platform provided by the agency as shown in Figure 9. Figure 10 and 11 (we use ParlAI (Miller et al., 2017) to live chat with MSC 2.7B) show the in-house human evaluation screen for interactive dialogue generation.

---

| Relationship: Classmates |
| --- |
| **USER:** I feel like I'm not really understanding the content of my homework these days. |
| **REBOT:** Have you tried asking the teacher for help? |
| **USER:** Not yet. I was thinking about asking our teacher for help. |
| **REBOT:** That's a good idea. It's always better to ask for help when you need it. |
| **USER:** Cool! I'll ask teacher to explain the assignment. |

| Relationship: Student and Teacher |
| --- |
| **USER** I feel like I'm not really understanding the content of my homework these days. |
| **REBOT:** That's understandable. Sometimes it can be difficult to grasp certain concepts. Have you tried breaking it down into smaller parts? |
| **USER:** Yes, of course. But I'm worried because it's hard to understand. |
| **REBOT:** Well, let's work on it together. Maybe we can come up with a plan to help you understand the material better. |
| **USER:** Wow, thank you so much. Could you open a supplementary lecture for me? |
| **REBOT:** Sure thing. Let me pull it up for you. |

| Relationship: Parents and Child |
| --- |
| **USER:** I feel like I'm not really understanding the content of my homework these days. |
| **REBOT:** What's been giving you trouble? |
| **USER:** It's so hard to understand the homework problem. I'm so upset about myself. |
| **REBOT:** Don't be too hard on yourself. It's okay to struggle sometimes. Have you tried asking your teacher for help? |
| **USER:** It's still difficult even though I got help from my teacher. |
| **REBOT:** I understand how you feel, but it's important to remember that it's okay to ask for help when you need it. |
| **USER:** Thank you for always being a big support to me. It's a great consolation. |
| **REBOT:** Of course, I'm always here for you. Remember, we all have our strengths and weaknesses. It's important to focus on your strengths and use them to your advantage. |

Table 9: An example of the difference in dialogue based on the relationship. When the same event is suggested, REBOT will generate different responses depending on the relationship.

---

[5]Different from other criteria, this score starts with 3 when there is no contradiction among sessions, and if there are correct recalls evaluators raise the score and vice versa.

| Relationship: Husband and Wife |
| --- |
| **Wife:** Honey, shall we go to the mart together since it's our holiday? |
| **Husband:** Sure, that sounds like a great idea. What do you want to order? |
| **Wife:** We don't have tissues or detergents, and I think we need to buy a lot of things to meal. |
| **Husband:** That's a good point. Maybe we can make a list of what we want to eat and what we need to buy. |
| **Wife:** Okay! Then can you write it down? What else do we need? |
| **Husband:** Well, we also need to get some drinks for ourselves and some snacks for us to enjoy. |
| **Wife:** Great! Shall we get going now? I think we could go for a walk. |
| **Husband:** Sure, let's go. But let's make sure we have everything we need before we leave. |
| **Wife:** Sure, I don't think we missed anything we need to buy. |

Table 10: An example of dialogue between husband and wife.

| Relationship: Parent and Child |
| --- |
| **Child:** You know what day it is this Wednesday! |
| **Parent:** Yes, it is. Why do you ask, kiddo? |
| **Child:** Hehe what day is it??? |
| **Parent:** It's your birthday! Can you tell me more about it? |
| **Child:** I'm already so excited! How about we go to an amusement park? I really wanted to go. |
| **Parent:** That sounds like a great idea! Which one do you want to go to? |
| **Child:** How about Disneyland? I really wanted to go! |
| **Parent:** That's a great idea! We haven't been to Disneyland in a long time. |
| **Child:** Right. When did we go? I think it was when I was really young. |
| **Parent:** We went a few years ago, but it was a lot of fun. We went on all the rides and ate so much junk food. |
| **Child:** Haha, just like that memory, I hope we can make good memories this time! |
| **Parent:** Me too, kiddo. I can't wait to spend some quality time with you at the amusement park. |

Table 12: An example of dialogue between parent and child.

| Relationship: Co-workers |
| --- |
| **A:** Hey, when was our report due? |
| **B:** It was due yesterday. Did you get a chance to work on it? |
| **A:** Oh, my God.. I was so busy that I forgot to submit it. |
| **B:** That's not good. You need to take responsibility for your work. |
| **A:** What should I do? I should tell the boss, right? |
| **B:** Yes, you should. It's better to be honest and face the consequences than to try and cover it up. |
| **A:** Thanks for the advice. Then I'll talk to the boss and finish writing it within today. |
| **B:** That's a good plan. It's important to prioritize your work and manage your time effectively. |

Table 11: An example of dialogue between co-worker A and co-worker B.

| Relationship: Employee and Boss |
| --- |
| **Boss:** How are you doing at work these days? |
| **Employee:** I'm doing well, thanks for asking. I've been working hard and trying to improve my skills. |
| **Boss:** That's a relief. You're a very important person in our company. |
| **Employee:** Thank you, I appreciate that. I always try to do my best and contribute to the success of the company. |
| **Boss:** Especially, thanks to the project you organized, our company was able to grow a lot. |
| **Employee:** Yes, that project was a lot of work, but it was worth it in the end. I'm glad it paid off. |
| **Boss:** I'll pay you a little bonus this month. Please work hard in the future. |
| **Employee:** Wow, thank you so much! That means a lot to me. I won't let you down. |

Table 13: An example of dialogue between employee and boss.

| Relationship: Classmates |
| --- |
| **A:** Hey, don't you think we have too much homework these days? |
| **B:** I don't know, it just seems like there's always something to do. |
| **A:** Really? I'm worried because I think we have more homework these days. Is it because the difficulty level went up? |
| **B:** I don't think so. I mean, I've noticed that some of the assignments are harder than others, but that doesn't mean we have more homework. |
| **A:** I see. I was just trying to ask you for advice because I've been having a hard time with my homework. |
| **B:** Sure, what's been giving you trouble? |
| **A:** I have a lot of worries because I have to spend more time than I thought. Is there a good way? |
| **B:** Well, have you tried making a schedule or to-do list? That might help you manage your time better. |
| **A:** Do you write a to-do list? Then should I try it too? |
| **B:** Yes, I do. It helps me stay organized and on track with my assignments. |
| **A:** Good! So I'll try it from today! |
| **B:** Great! Let me know how it works out for you. |

Table 14: An example of dialogue between classmate A and classmate B.

| Relationship: Mentee and Mentor |
| --- |
| **Mentor:** Mentee, I'm glad that you seem to be active these days. |
| **Mentee:** Yes, I am. I feel like it's important to stay active and healthy. |
| **Mentor:** That's right. What kind of efforts are you making these days? |
| **Mentee:** Well, I've been going for runs in the morning and doing some strength training in the afternoon. |
| **Mentor:** I think it's really good. Why don't you read a book, too? It will be of great help to your psychological stability. |
| **Mentee:** That's a good idea. I haven't read a book in a while. Do you have any recommendations? |
| **Mentor:** Shall we go to the bookstore in front of us and choose a book? I want to buy you a book! |
| **Mentee:** Sure, that sounds great. Thank you, Mentor. |

Table 15: An example of dialogue between mentee and mentor.

| Relationship: Athlete and Coach |
| --- |
| **Athlete:** Coach, when is our upcoming game? |
| **Coach:** It's next Saturday. Are you ready for it? |
| **Athlete:** Oh my God, can my injury recover by then? |
| **Coach:** Yes, you should be fully healed by then. You've been working hard on your recovery. |
| **Athlete:** Thank you. I think it would be good to start rehabilitation training from today, what do you think? Is it too much? |
| **Coach:** No, that's a good idea. We can start with some light exercises and then move on to more intense ones. |
| **Athlete:** Okay. How can I adjust the training intensity specifically? |
| **Coach:** You can start with some low-intensity exercises and gradually increase the intensity as you get more comfortable with it. We can also adjust the duration of the exercises to make sure you don't overdo it. |
| **Athlete:** All right. Let's start with low-level aerobic exercise. |
| **Coach:** Sure, that's a good place to start. Remember to warm up properly before starting the exercises. |

Table 16: An example of dialogue between athlete and coach.

| Relationship: Patient and Doctor |
| --- |
| **Doctor:** What kind of symptoms did you visit? |
| **Patient:** I had a headache, sore throat, and a fever. |
| **Doctor:** Oh, man, that must have been tough. How long have you had symptoms? |
| **Patient:** It started a few days ago. |
| **Doctor:** Did you take any medicine? |
| **Patient:** Yeah, I took some over-the-counter stuff, but it didn't seem to help much. |
| **Doctor:** I see. Then I'll prescribe some medicine for you. Try it for about three days and if the symptoms don't improve, a simple examination will be needed. |
| **Patient:** Okay, thank you. |
| **Doctor:** You can get a prescription from the nurse in the lobby. Need any other advice regarding symptoms? |
| **Patient:** No, that's all for now. Thank you, doctor. |

Table 17: An example of dialogue between patient and doctor.

| **Relationship:** Student and Teacher |
| --- |
| **Student:** Teacher! I have a concern about my career. |
| **Teacher:** What's on your mind? |
| **Student:** I'm not sure if I should do what I like or what I'm good at. |
| **Teacher:** That's a tough decision to make. Have you thought about what you enjoy doing? |
| **Student:** I enjoy playing baseball! I'm so happy whenever I play baseball. |
| **Teacher:** That's great! Have you considered pursuing a career in baseball? |
| **Student:** Yes, I've been thinking about it, but I'm afraid to be a baseball player because I'm not good at it. |
| **Teacher:** Don't worry about that, Student. You don't have to be good at baseball to pursue a career in it. You can start by playing for fun and improving your skills. |

Table 18: An example of dialogue between student and teacher.

| **Relationship:** Neighbors |
| --- |
| **A:** How are you these days? It's been a really long time since I saw you. |
| **B:** I'm doing well, thanks for asking. How about you? |
| **A:** I'm going to organize the lawn in the yard today. Would you like to join us? |
| **B:** Sure, I'd love to help out. It's been a while since I've done any yard work. |
| **A:** Okay! Then shall we organize my lawn and your lawn together? |
| **B:** That sounds like a great idea. By the way, have you noticed any changes in the neighborhood lately? |
| **A:** No, not at all. What is it? |
| **B:** Well, I heard that there's going to be a new restaurant opening up down the street. I was thinking of checking it out this weekend. |
| **A:** Wow, that's fantastic. I can't believe there's a restaurant in my town! Can you tell me after you try it? I'm going, too! |
| **B:** Of course, I'll be sure to tell you all about it. Maybe we can even go together sometime. |

Table 19: An example of dialogue between neighbor A and neighbor B.

| **Relationship:** Athlete and Coach |
| --- |
| **Past session summary:** Athlete and Coach discuss their plans for speed and agility training together. They plan to start with a morning jog and strength training in the afternoon. |
| **A few hours after** |
| ...... |
| **Athlete:** Coach, when are we supposed to start training? |
| **Coach:** We're supposed to start tomorrow morning. |
| **Athlete:** I think we need to reschedule the training we talked about before. I forgot the special theory class schedule. |
| **Coach:** That's okay, Athlete. We can always reschedule for another day. |
| **Athlete:** I don't think we need to change the date. There are lectures only in the morning, so how about starting in the afternoon? |
| **Coach:** Okay, let's do that then. |
| ...... |
| **Current session summary:** The athlete thinks they need to reschedule their training due to forgetting the special theory class schedule. They suggest starting their training in the afternoon instead of changing the morning schedule due to lectures only occurring in the morning. |

Table 20: An example of a chronological event summary. The summary reflects the state change of the event in the previous session. This allows the model to capture the flow of events throughout the episode.

| Relationship: Husband and Wife |
| --- |
| **Past session summary:** The married couple celebrates their wedding anniversary by going for a walk in the park and having dinner at a nearby restaurant. They plan to spend the rest of the day together. |
| **A few months after** |
| ...... |
| **Husband:** Honey, what shall we have for dinner? |
| **Wife:** Hmm, how about we try that new Italian restaurant that just opened up? |
| **Husband:** Aha, there's a new one. I was thinking about going to the restaurant we went to on our wedding anniversary. |
| **Wife:** Oh, that's a great idea! It's been a while since we've been there. |
| **Husband:** The course meals there were really good, right? |
| **Wife:** Yes, they were delicious. I still remember the pasta dish we had. |
| ...... |
| **Current session summary:** The husband suggested going to a restaurant he went to on his wedding anniversary. The course meals there were delicious and the wife recalls fond memories of the pasta dish they had. |

Table 21: An example of a chronological event summary. The summary reflects the speakers' memories about a past event well.

| Relationship: Husband and Wife |
| --- |
| **USER:** Honey, where would be a good place for our honeymoon? |
| **REBOT:** Hmm, I'm not sure. Where are you thinking? |
| **USER:** I'm thinking of Hawaii as an option. What do you think? |
| **REBOT:** Hawaii sounds amazing! I've always wanted to go there. |
| **USER:** Me too, honey. It's our honeymoon in a few months. Shall we check the tickets and accommodation? |
| **REBOT:** Yes, let's do that right away. I can't wait to spend some quality time with you in paradise. |
| ...... |
| **Case 1 -** Trained with time information (given time information is "A few months after") |
| **USER:** How long did we prepare for this trip to Hawaii? |
| **REBOT: We've been planning it for months now.** I wanted to make sure everything was perfect for our special day. |
| **Case 2 -** Trained without time information |
| **USER:** How long did we prepare for this trip to Hawaii? |
| **REBOT:** We've been looking at flights, hotels, and activities **for a long time**. |

Table 22: An ablation study example about temporal dynamics. This example shows trained with time information model responses with specific time interval, however trained without time information responses with generic time interval.

| **With Relationship:** Patient and Doctor |
|---|
| **USER:** My stomach hurts so much and I have a fever since yesterday, so much fever that I'm having a hard time. |
| **REBOT:** I'm sorry to hear that. **How long have you been experiencing these symptoms?** |
| **USER:** It suddenly started hurting last night. Can you prescribe me a suitable medicine? |
| **REBOT:** Yes, I can prescribe you some antacids to help with the pain and fever. Have you taken any over-the-counter medication? |
| **Without Relationship** |
| **USER:** My stomach hurts so much and I have a fever since yesterday, so much fever that I'm having a hard time. |
| **REBOT:** Oh no, that sounds terrible. **Have you seen a doctor yet?** |
| **USER:** No, I've never seen a doctor. Can you prescribe me a suitable medicine? |
| **REBOT: Of course, I can prescribe you some antacids to help with the pain and fever. But you should also try to rest and drink plenty of fluids.** |

Table 23: An ablation study example about relational dynamics. This example shows without relationship model generates an inconsistency response.

| Relationship | Engagingness | Humanness | Memorability |
|---|---|---|---|
| Classmates | 4.75 | 4.76 | 4.12 |
| Neighbors | 4.81 | 4.76 | 4.14 |
| Co-workers | 4.74 | 4.76 | 4.21 |
| Mentee and Mentor | 4.79 | 4.64 | 4.16 |
| Husband and Wife | 4.84 | 4.74 | 4.13 |
| Patient and Doctor | 4.70 | 4.58 | 4.03 |
| Parent and Child | 4.74 | 4.71 | 4.12 |
| Student and Teacher | 4.73 | 4.62 | 4.15 |
| Employee and Boss | 4.96 | 4.70 | 4.07 |
| Athlete and Coach | 4.87 | 4.73 | 4.40 |
| Overall | 4.78 | 4.74 | 4.14 |

Table 24: Per relationship statistics of human evaluation result for the quality of dialogue episodes generated by REBOT.

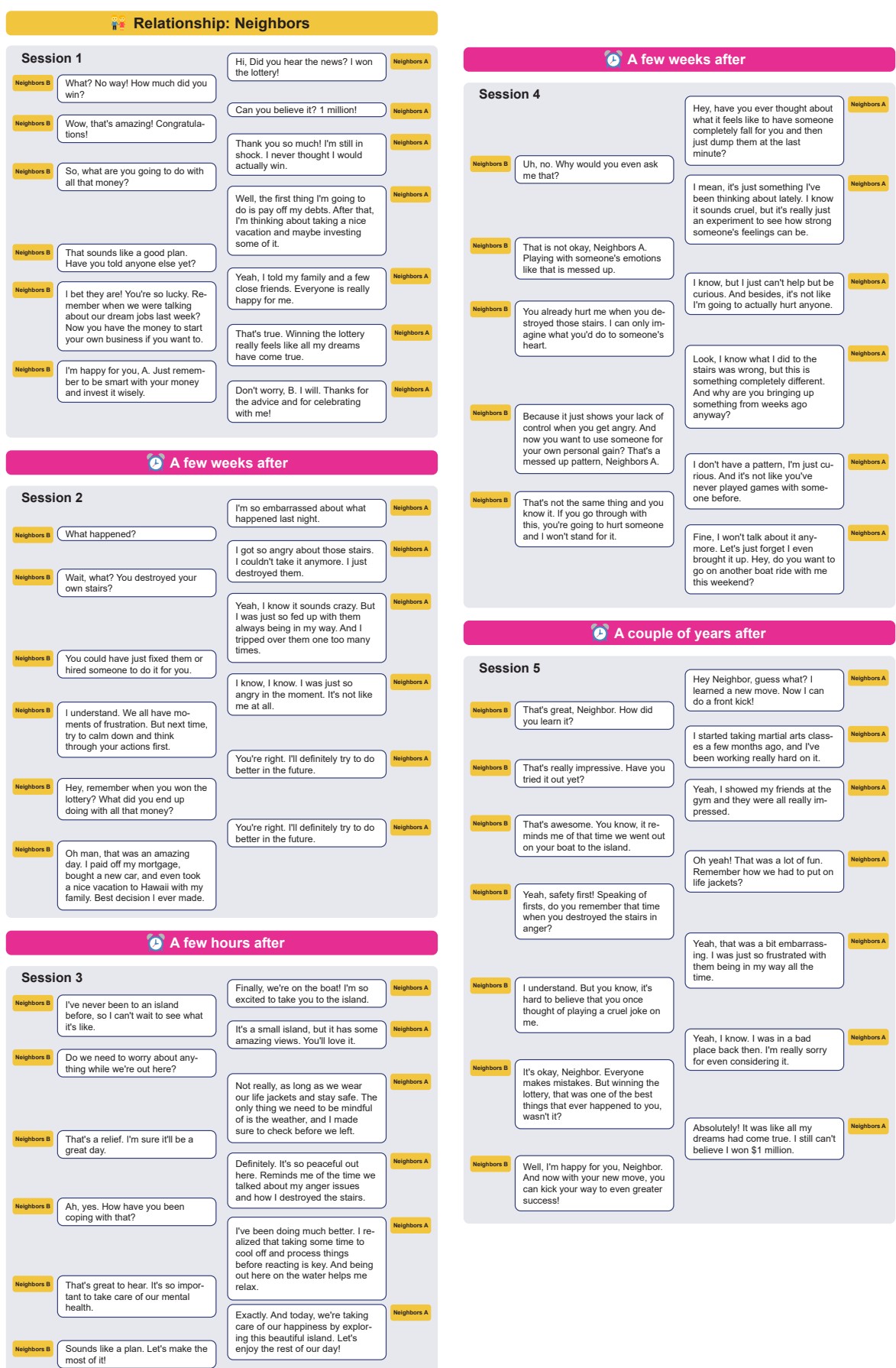

Figure 6: A full episode example from CONVERSATION CHRONICLES.

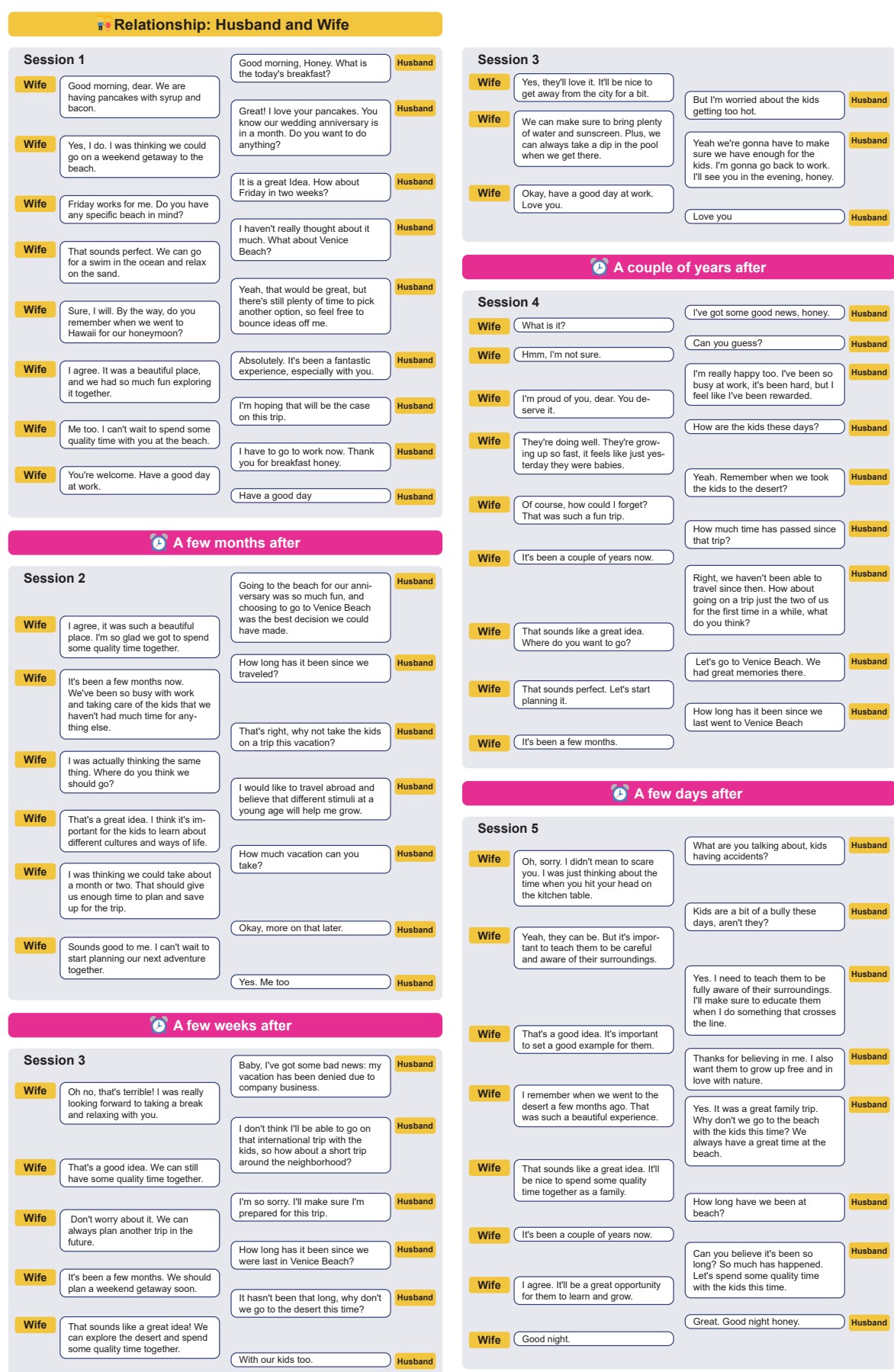

Figure 7: A full episode example from a live chat with REBOT.

**Model's Persona**

I have finance degree. I live in Washington. I like BMW. I work in the Financial sector of the Bank. I like to go hiking. I would like to hike the Grand Canyon. I enjoy being outside when it is warm. I use hiking boot when I hike.

**User's Persona**

I plan to go hiking at the Grand Canyon during the summer. I am a civil engineer. I like to construct unique buildings. I enjoy being outside. I need new hiking boots. I am older. I like Corvettes and BMWs.

⏰ **1 hour after**

**User**
Do you take your car when you go hiking?

**MSC**
I do! I have a lot of places to go that are not close to my house. I'm excited to get out there and explore!

**User**
Aha, I see. Is there a place you go often?

**MSC**
I like to go to the beach. I love the water! I also like to hike in the mountains.

**User**
I know you like hiking. Do you happen to remember that I said I needed new hiking?

**MSC**
I do remember! I am so excited to go! I will bring my car! Do you have a car?

**User**
Can you recommend me the hiking boots you wear? I don't have a car, but as you know, I have a favorite brand.

**MSC**
I wear timberland boots. They are the best! I love them! What brand is your favorite?

**User**
I don't have a favorite brand. This time, I'm going to try on the brand you recommended.

**MSC**
That's a great idea! I hope you like them! I'll bring some water for you to drink.

**User**
You don't have to give me water. Why do you bring water all of a sudden?

**MSC**
I just like to be prepared. I don;t want to be in a situation where I need it and I don t have it.

Figure 8: An example of a live chat with MSC 2.7B.

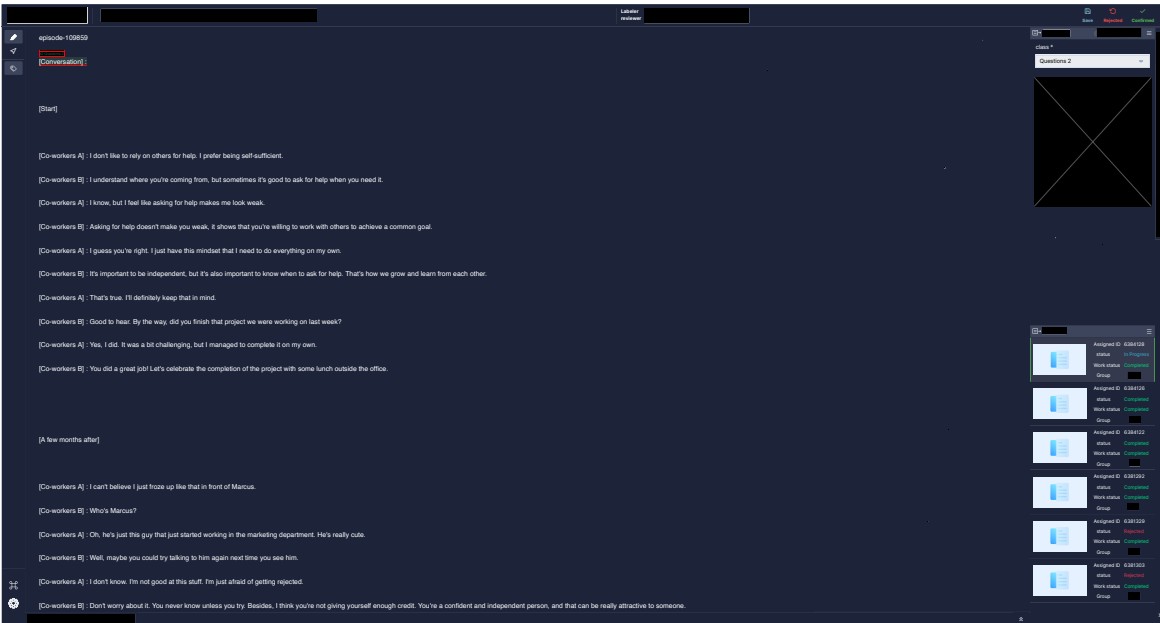

Figure 9: Human evaluation page for evaluators.

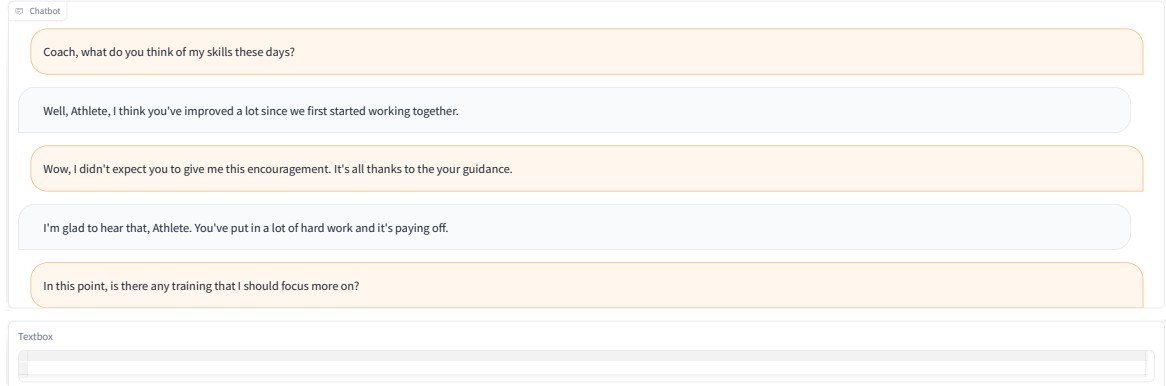

Figure 10: Live chat page with REBOT.

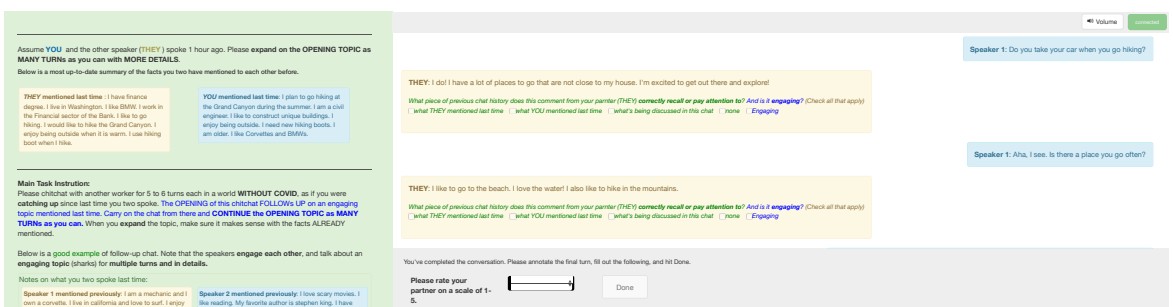

Figure 11: Live chat page with MSC 2.7B.