# OpenReview forum: "Conversation Chronicles: Towards Diverse Temporal and Relational Dynamics in Multi-Session Conversations"
_EMNLP/2023/Conference — EMNLP 2023 Main_

### Official Review · Reviewer_xS68 · 2023-08-05

**Soundness:** 4

**Excitement:**

4: Strong: This paper deepens the understanding of some phenomenon or lowers the barriers to an existing research direction.

**Paper Topic And Main Contributions:**

This work explores the direction in open-domain conversation where we have multi-session conversation history and various timestamps (from hours to years). To help support consuming conversation history from previous sessions, the authors collected a large-scale dataset, CONVERSATION CHRONICLES, that contains multi-session conversations from parties with various relationships and time intervals between sessions. The authors also proposed a model that shows good human evaluation result.

**Questions For The Authors:**

1. As I mentioned above, s there any mechanism of quality control or verification in LLM (i.e. ChatGPT) outputs?

**Reasons To Accept:**

The main reasons to accept:

1. There is little work done on multi-session conversation history in open-domain conversion. The dataset proposed in this work could be very helpful for further studies.
2. Dataset is carefully collected in large-scale and it is interesting to see another case where LLM can help reduce the cost of dataset collection.
3. Paper is well written and presented.

**Reasons To Reject:**

The main reasons to reject:

Some details are missing:
1. Despite the fact that LLM is powerful at generation, we should still be careful about the actual outputs. Is there any mechanism of quality control or verification in LLM (i.e. ChatGPT) outputs?
2. Not clear about the model performance. Although the section 5.6 mentions that MSC 2.7B is used for comparison, there there is no human evaluation reported for MSC 2.7B.

**Reproducibility:**

4: Could mostly reproduce the results, but there may be some variation because of sample variance or minor variations in their interpretation of the protocol or method.

**Reviewer Confidence:**

3: Pretty sure, but there's a chance I missed something. Although I have a good feel for this area in general, I did not carefully check the paper's details, e.g., the math, experimental design, or novelty.

---

> ### Author Rebuttal · Authors · 2023-08-29
>
> Thank you for your valuable feedback on our paper. Please find our responses below.
>
> **(1) Ensuring quality of data from LLMs.**
>
> After generating data using the LLMs, we implemented an automatic filtering process. This process eliminates outputs that do not satisfy specific criteria, such as those containing harmful content and corner cases (appearance of unrelated speakers, stage directions, etc.). We will include the automatic methods we used for data filtering in the paper.
>
> However, relying solely on automated methods for quality verification has its limitations. To complement these methods, we conducted extensive human evaluations—these evaluations encompassed around six thousand episodes, a scale far surpassing that of previous studies which typically involve several hundred episodes—based on various criteria, including consistency, coherence, time interval, and relationship.
>
> While we might not be able to eliminate all noisy data, our extensive human evaluation yielded high scores, as shown in Table 4 and Figure 3 (ours vs. human-generated dataset). This strengthens our confidence in the dataset's quality. Thus, through a combination of automated filtering and large-scale human evaluations, we believe our dataset meets high-quality standards.
>
> **(2) Human evaluation result for MSC 2.7B.**
>
> The human evaluation results comparing ReBot and MSC 2.7B are reported in Figure 5 in the ‘Interactive Dialogue Generation’ paragraph of Section 6. We have explained all experiments and evaluation details in Section 5. Following that, Section 6 presents all the experimental and evaluation results (the only exception is the dataset evaluation results being presented in Section 3.4). We will make necessary update to ensure these details stand out for easier navigation and understanding.

---

### Official Review · Reviewer_Eydv · 2023-08-06

**Soundness:** 3

**Excitement:**

3: Ambivalent: It has merits (e.g., it reports state-of-the-art results, the idea is nice), but there are key weaknesses (e.g., it describes incremental work), and it can significantly benefit from another round of revision. However, I won't object to accepting it if my co-reviewers champion it.

**Paper Topic And Main Contributions:**

The paper addressed the problem of multi-session dialogue generation. It provides a new LLM-generated corpus of multi-session chats and shows they are of better quality than the previous human-generated data. The paper then proposed a new model to generate responses and summarize chats.

**Reasons To Accept:**

1. The paper addressed an important problem in dialogue system and would be of interest to EMNLP audience
2. The paper provided a new machine-generated corpus that is provably better than human-generated corpus in certain criteria

**Reasons To Reject:**

1. The paper contains modeling work only at a bare minimum. The paper would be stronger if it positions itself as a resource paper, and includes more analysis on data generation. For example, it would be interesting to see the impact of prompt engineering on the generated corpus.
2. In the post-processing of the LLM-generated corpus, the human curation part seems weak. There is no common sense filtering, nor did the authors evaluate the corpus based on common sense. So the gap between this corpus and human-generated data is unclear.


**Reproducibility:**

4: Could mostly reproduce the results, but there may be some variation because of sample variance or minor variations in their interpretation of the protocol or method.

**Reviewer Confidence:**

3: Pretty sure, but there's a chance I missed something. Although I have a good feel for this area in general, I did not carefully check the paper's details, e.g., the math, experimental design, or novelty.

---

> ### Author Rebuttal · Authors · 2023-08-29
>
> Thank you for your valuable feedback on our paper. Please find our responses below.
>
> **(1) Data generation.**
>
> The primary focus of our study is the quality of the generated dialogues rather than the data generation process itself. While we explored many options before finalizing the process, we did not present all the options in the paper (we added the full prompt in the appendix for reference though) since previous research has already shown that well-crafted prompts can effectively distill knowledge from LLMs. Instead, our attention has been directed more toward evaluating the quality of generated dialogues through comprehensive human evaluations. That said, we will add more prompts that we tried but did not work best in the paper for the benefit of the research community. Thanks for the suggestion.
>
> **(2) Commonsense.**
>
> We would like to clarify that our argument is not that our dialogue dataset contains common sense. Nonetheless, if we interpret ‘common sense’ in a broad context (i.e., general knowledge), we believe that the dialogues in our dataset have been verified to have enough common sense to hold natural conversation through extensive human evaluation on various criteria; without such common sense, the dialogues would likely receive lower scores (Table 4). And, as shown in Figure 3, our dataset can be considered comparable with human-generated data. When trained on the dataset, our dialogue model exhibits high performance on evaluation metrics such as ‘Engagingness’ and ‘Humanness’. This suggests that the model contains a sufficient amount of general common sense for conversation, as shown in Table 5.

---

### Official Review · Reviewer_fauZ · 2023-08-12

**Typos Grammar Style And Presentation Improvements:** Although Figure 1 is clear, This is n…
**Soundness:** 4

**Excitement:**

4: Strong: This paper deepens the understanding of some phenomenon or lowers the barriers to an existing research direction.

**Paper Topic And Main Contributions:**

This paper presents a novel multi-session dialogue dataset called Conversation Chronicles. Furthermore, the authors have incorporated various time intervals and fine-grained speaker relationships to facilitate long-term conversation modeling. The dataset's quality is superior, boasting a significantly higher number of episodes compared to previous multi-session conversation datasets.

**Reasons To Accept:**

1. This paper is well-written and easy to follow.


2. The authors have constructed multi-session dialogues based on 10 pre-defined relationships between speakers, which presents a novel and intriguing approach.

3. An extensive human evaluation was undertaken to gauge the quality of both the dataset and the generated dialogues. Consequently, the Conversation Chronicles dataset and the ReBot outperformed the MSC dataset and the MSC 2.7 model, respectively.

**Reasons To Reject:**

1. There seems to be a gap in empirical experimentation. Given that each multi-session dialogue is mapped to one of the 10 predefined relationships, a detailed evaluation of dialogues generated for each specific relationship would enhance the empirical robustness of the study.


2. How might one measure the impact of speaker relationships on the multi-session conversation dataset? Specifically, if the dataset did not factor in speaker relationships, would there be a noticeable performance drop in ReBot or other long-term conversation models?


3. Do the authors operate under the assumption that these relationships remain static throughout multi-session conversations? I'm curious as to why potential shifts in relationships weren't considered. For instance, in a "Co-workers" scenario, if one or both individuals retire (or quit), they no longer share the same professional bond. In my opinion, in real-life contexts, relationships often evolve, especially over extended periods, such as several years.


4. In the comparison between ReBot and MSC 2.7B, do the authors consider this comparison to be fair? There appears to be a significant discrepancy in the number of sessions between the two. Could the enhanced performance of ReBot be attributed to the more extensive training data it had access to?

**Reproducibility:**

3: Could reproduce the results with some difficulty. The settings of parameters are underspecified or subjectively determined; the training/evaluation data are not widely available.

**Reviewer Confidence:**

4: Quite sure. I tried to check the important points carefully. It's unlikely, though conceivable, that I missed something that should affect my ratings.

---

> ### Author Rebuttal · Authors · 2023-08-29
>
> Thank you for your valuable feedback on our paper. Please find our responses below.
>
> **(1) Evaluation of relationship.**
>
> We verified the fine-grained relationships in our dataset using extensive human evaluation. The results, as presented in Table 4, indicate high scores, suggesting that the dataset accurately reflects the speaker relationships. Our dialogue model, ReBot, was trained on this dataset. Consequently, we anticipate that ReBot can recognize and internalize these speaker relationships. To validate ReBot's capability in this regard, we conducted another human evaluation. Here, evaluators were informed of the speaker relationships and then tasked with assessing the quality of the dialogues generated by ReBot based on various criteria. These results are detailed in Table 5 and Figure 5. Given that the evaluators were cognizant of the relationships, the high scores from this evaluation reaffirm that ReBot's generated dialogues effectively mirror the intended speaker relationships. For further clarity, we've included dialogue examples for each relationship in the Appendix (Table 9 to 18, showcasing how these relationships influence dialogue generation).
>
> **(2) Model performance without relationship and time interval.**
>
> Thank you for pointing that out. We conducted an ablation experiment to assess the model's performance when trained with and without relationship information. The results showed that, when trained without relationship context, the model does not maintain a consistent relationship with a user. In contrast, when trained with relationship information, the model demonstrates a smooth and consistent conversation flow.
>
> Please consider this example:
>
> > **With Relationship** (doctor and patient):
> USER: My stomach hurts so much and I have a fever since yesterday, so much fever that I'm having a hard time.
> ReBot: I’m sorry to hear that. **How long have you been experiencing these symptoms?**
> USER: It suddenly started hurting last night. Can you prescribe me a suitable medicine?
> ReBot: Yes, I can prescribe you some antacids to help with the pain and fever. Have you taken any over-the-counter medication?
>
> > **Without Relationship**:
> USER: My stomach hurts so much and I have a fever since yesterday, so much fever that I'm having a hard time.
> ReBot: Oh no, that sounds terrible. **Have you seen a doctor yet?**
> USER: No, I've never seen a doctor. Can you prescribe me a suitable medicine?
> ReBot: **Of course, I can prescribe you some antacids to help with the pain and fever. But you should also try to rest and drink plenty of fluids.**
>
> As evident from the 'Without Relationship' dialogue, the model’s responses, particularly **"Have you seen a doctor yet?"** and **"Of course, I can prescribe …"**, are inconsistent regarding its role.
>
> Additionally, we also conducted an ablation study to assess the model's performance when trained with and without time information.
>
> Please consider this example:
>
> > USER: Honey, where would be a good place for our honeymoon?
> ReBot: Hmm, I'm not sure. Where are you thinking?
> USER: I'm thinking of Hawaii as an option. What do you think?
> ReBot: Hawaii sounds amazing! I've always wanted to go there.
> USER: Me too, honey. It's our honeymoon in a few months. Shall we check the tickets and accommodation?
> ReBot: Yes, let's do that right away. I can't wait to spend some quality time with you in paradise.
> …..
>
> > **Trained with time information** (and given a time info. **'A few months after'** when generating response):
> …..
> USER: How long did we prepare for this trip to Hawaii?
> ReBot: **We've been planning it for months now**. I wanted to make sure everything was perfect for our special day.
>
> > **Trained without time information:**
> …..
> USER: How long did we prepare for this trip to Hawaii?
> ReBot: We've been looking at flights, hotels, and activities **for a long time**.
>
> As shown in the “without time information” response, the model generates generic time information (**"for a long time”**) instead of a specific time amount.
>
> This time, we provide time interval to the model trained without time information. Here are the comparisons.
>
> > USER: Mom! Let's go to an amusement park this weekend!
> ReBot: That sounds like a great idea! Which one do you want to go to?
> USER: How about Disneyland?
> ReBot: Oh, I've always wanted to go to Disneyland! Let's do it!
> …..
>
> > **Trained with time information** (Time interval: **'A few hours after'**):
> USER: When did we talk about amusement parks?
> ReBot: **We talked about it earlier today.** Do you remember?
>
> > **Trained without time information** (Time interval: **'A few hours after'**):
> USER: When did we talk about amusement parks?
> ReBot: **We talked about it a few weeks ago** when I asked you if you wanted to go to one with me.
>
> As shown in the response, the model responds with incorrect time information (**“a few weeks ago”**).
>
> We will update the paper with these ablation experiment results.
>
> **(3) Shifts in relationships.**
>
> Thank you for the suggestion. Our primary objective in this study is to determine whether we can develop a dialogue model that consistently yet dynamically establishes relationships between speakers over a series of dialogue sessions. Accordingly, our model is also equipped to handle shifts in relationships due to events and the passage of time, as our dataset incorporates both temporal and relational dynamics. Below is an example of a model-user interaction:
>
> > **[Relationship: Athlete and Coach]**
> USER: Actually, I've been thinking about it since my last injury, but now I'm going to retire as an athlete and live a new life.
> .....
> USER: Thank you. I will probably take coaching training for a year, and I am planning to be a full-time coach after that. It would be meaningful to meet each other as a coach later on.
> ReBot: That sounds like a great plan. I wish you all the best in your coaching career, Athlete.
> .....
> **[A couple of years after]**
> USER: Hey, I finished my training course and finally became a full-time coach like you!
> ReBot: That's great news! I knew you had it in you.
> USER: **Thank you! As I became an official coach, I'd like to learn what I need to know as a coach, could you tell me?**
> …..
> ReBot: **Exactly. As a coach, you need to understand your players and their motivations. You need to be able to connect with them on a deeper level and help them reach their full potential.**
> USER: Thank you! I will be a coach who can raise the potential of the players like you!
> ReBot: **I have no doubt that you will. Just remember to stay true to yourself and your values as a coach, and always lead by example.**
>
> As illustrated in the example, the initial relationship was that of an 'Athlete and Coach'. Athlete decided to be a coach, transitioning to work alongside their former coach as peers. The model can recognize the change in relationship and respond to the conversation accordingly.
>
> We will update the paper with this dynamic relationship shift analysis.
>
> **(4) Comparison with MSC 2.7B.**
>
> Technically, the model architectures of ReBot and MSC 2.7B differ. Unlike MSC 2.7B, ReBot is designed to take both the speaker relationship and time interval as inputs. This makes a direct comparison challenging. However, our rationale for selecting MSC 2.7B as a benchmark is its distinction as the only publicly available model designed for multi-session conversations. If MSC 2.7B were modified to accept speaker relationship and time interval inputs and then trained on our dataset, it might achieve performance comparable to ReBot. However, it is difficult to consider the dataset and model architecture separately.
>
> Thus, we believe that our contributions—developing a large-scale dataset incorporating speaker relationships and time intervals, and a model design that uniquely leverages these factors—collectively enable such a natural and engaging conversational experience.
>
> **(5) About Figure 1.**
>
> Thank you for pointing that out. We should have mentioned Figure 1 at line 084 in Section 1, but we missed it. We will update this in the paper.

---

### Meta-Review · Area_Chair_KwUZ · 2023-10-06

**Recommendation:** 5

**Metareview:**

The paper presents a significant contribution to the field of dialogue systems by introducing a novel approach to constructing multi-session dialogues and providing a machine-generated corpus. The resources and insights offered in this paper can be expected to facilitate future studies in this area.

---

### Decision · Program_Chairs · 2023-10-07

**Decision:**

Accept-Main

**Comment:**

The paper presents a significant contribution to the field of dialogue systems by introducing a novel approach to constructing multi-session dialogues and providing a machine-generated corpus. The resources and insights offered in this paper can be expected to facilitate future studies in this area.